# Persistent Transmission of HCV among Men Who Have Sex with Men despite Widespread Screening and Treatment with Direct-Acting Antivirals

**DOI:** 10.3390/v14091953

**Published:** 2022-09-02

**Authors:** Stephanie Popping, Lize Cuypers, Mark A. A. Claassen, Guido E. van den Berk, Anja De Weggheleire, Joop E. Arends, Anne Boerekamps, Richard Molenkamp, Marion P. G. Koopmans, Annelies Verbon, Charles A. B. Boucher, Bart Rijnders, David A. M. C. van de Vijver

**Affiliations:** 1Department of Viroscience, Erasmus Medical Center, 3015 CA Rotterdam, The Netherlands; 2Department of Medical Microbiology and Infectious Diseases, Erasmus Medical Center, 3015 CA Rotterdam, The Netherlands; 3Laboratory of Clinical and Epidemiological Virology, Department of Microbiology, Immunology and Transplantation, Rega Institute for Medical Research, KU Leuven, 3000 Leuven, Belgium; 4Department of Laboratory Medicine, University Hospitals Leuven, 3000 Leuven, Belgium; 5Department of Internal Medicine and Infectious Diseases, Rijnstate Ziekenhuis, 6815 AD Arnhem, The Netherlands; 6Department of Internal Medicine and Infectious Diseases, OLVG, 1091 AC Amsterdam, The Netherlands; 7Department of Clinical Science, Institute of Tropical Medicine Antwerp, 2000 Antwerp, Belgium; 8Department of Internal Medicine and Infectious Diseases, Universitair Medisch Centrum Utrecht, Utrecht University, 3584 CX Utrecht, The Netherlands

**Keywords:** hepatitis c, transmission dynamics, phylogenetic analysis, HIV-infected Men-who-have-sex-with-men, hepatitis C elimination

## Abstract

Background: In the Netherlands, unrestricted access to direct-acting antivirals (DAAs) halved the incidence of acute hepatitis C virus (HCV) infections among HIV-infected men who have sex with men (MSM). To develop strategies that can further reduce the spread of HCV, it is important to understand the transmission dynamics of HCV. We used phylogenetic analysis of a dense sample of MSM to provide insight into the impact of unrestricted access to DAAs on HCV transmission in the Netherlands and in Belgium. Methods: We included 89 MSM that were recently infected with HCV genotype 1a in ten Dutch and one Belgian HIV treatment centers. Sequences were generated using next gene sequencing and Sanger sequencing. Maximum likelihood phylogenetic analysis (general time reversible model) was performed on concatenated NS5A and NS5B sequences and a reference set of 389 highly similar control sequences selected from GenBank. A cluster was based on a minimum bootstrap support of 90% and a 3% genetic distance threshold. Results: We found that 78 (88%) of individuals were part of seven major clusters. All clusters included individuals from across the study region, however, different cities were part of different clusters. In three clusters, HIV-negative MSM clustered with sequences from HIV-positive MSM. All clusters that were observed before the introduction of DAAs persisted after unrestricted access to DAAs became available. Conclusion: Recently acquired HCV infections among MSM in the Netherlands and Belgium are strongly clustered and therefore highly suitable for targeted prevention strategies, such as contact tracing and partner notification. Importantly, despite an HCV incidence reduction after high DAA uptake and continuously monitoring, HCV transmission persisted in the same clusters.

## 1. Introduction

The World Health Organization (WHO) set the goal to eliminate hepatitis C virus (HCV) as a global health threat by 2030. The WHO wants to achieve this ambitious goal by reducing the number of new HCV infections by 90% and by reducing HCV-related mortality by 65% in 2030 [1]. A key element of the WHO’s strategy of elimination is the expansion of HCV treatment with highly curative direct-acting antivirals (DAA) and thereby preventing onward transmission to others.

A recommended strategy to reach the goals set by the WHO is by a micro-elimination approach: the local scale-up of identifying new infections and DAA treatment in well-defined populations that actively contribute to the HCV epidemic. An effort to achieve micro-elimination was made in the Netherlands among HIV-infected men who have sex with men (MSM), which is the predominant Dutch population in whom HCV is actively transmitted. The micro-elimination approach included HCV testing every six months followed by immediate DAA treatment in HCV-infected people [2,3]. Although this effort of testing and expanded treatment access strongly reduced the number of acute infections by 50%, micro-elimination was not reached, as HCV continues to be transmitted [4].

Several Western European countries report high numbers of new HCV infections and re-infections among HIV co-infected MSM, which are mainly driven by ongoing engagement in high-risk behaviour [4,5,6]. Moreover, more reports mentioned the presence of HCV among HIV-negative MSM [7,8,9]. In addition to local transmission, HCV infections can be imported or acquired elsewhere and continue to seed the local epidemic, since viruses do not respect national borders [10]. Therefore, a better understanding of the effect of unrestricted DAA therapy on the transmission of recently acquired HCV among MSM is needed to select appropriate future interventions and to improve micro-elimination strategies.

In this study we use phylogenetic analysis to provide further insight into HCV transmission among mostly HIV-infected MSM with a HCV infection acquired before and after DAAs became unrestrictedly available: in the Netherlands and Belgium since 2015 and 2017, respectively. Similar as to the Netherlands the HCV epidemic in Belgium is mainly concentrated among MSM. 

## 2. Methods

### 2.1. Study Population of Acutely Infected HCV Genotype 1a Individuals 

Plasma samples from MSM with recently acquired HCV genotype 1a infections who participated in two Dutch Acute HCV in HIV studies (DAHHS) were collected between 2013–2014 and 2016–2018. Both DAHHS studies were single-arm, open-label, multicenter studies, investigating the efficacy of DAA therapies in mostly HIV-positive MSM with a recently acquired HCV infection [11,12]. A detailed description of the studies and patient inclusion criteria can be found elsewhere [11,12]. In summary, both DAHHS studies enrolled participants identified with an acute HCV infection during routine clinical care in ten different Dutch HIV treatment centers. Additionally, during the DAHHS 2 study, one Belgian HIV treatment center (Antwerp) participated, and patients from all over Belgium were referred to this center for participation to the study. In both studies, patients were included if HCV was diagnosed within six months after HCV infection, as described elsewhere [11,12]. All patients gave written informed consent for the use of their blood samples for research purposes [11,12]. 

The first DAHHS study started inclusions in September 2013 and lasted until December 2014. HCV genotype 1a acutely infected HIV-positive MSM (n = 57) were treated with boceprevir, pegylated interferon, and ribavirin for 12 weeks at a time when interferon-free treatment was yet unavailable [11]. 

The DAHHS 2 study started inclusion in February 2016 and ended in March 2018, where the impact of grazoprevir and elbasvir was studied for 8 weeks among 86 MSM acutely infected with HCV. Among those, 51 had a genotype 1a infection. Most of the patients were HIV-positive MSM, however, the study also included a small number (n = 4) of HIV-negative MSM [12]. 

### 2.2. Viral Sequencing of the Non-Structural Proteins 5A and 5B 

Two methods for sequencing were used in the two DAHHS studies, respectively: next-generation sequencing (NGS) in DAHHS 1 and Sanger population sequencing in DAHHS 2 [11,12]. In short, DAHHS 1 samples were sequenced by NGS on the Illumina MiSeq platform, as previously published [13]. The CLC Genomics Workbench (version 7.5/7.5.1) including the CLC Microbial Genome Finishing Module (version 1.4, Qiagen, Hilden, Germany), was used to process the raw reads into full genome consensus sequences. Trimmed reads were pre-filtered against a GenBank reference list containing 953 partial and complete HCV genomes, followed by the sampling of a subset of 50,000 to 100,000 HCV-specific sequence reads for the de novo assembly of whole genomes [13]. For subsequent phylogenetic and resistance analysis, only the nucleotide sequences of the NS5A and NS5B region were used. For the DAHHS 2 samples, solely the NS5A and NS5B genes were amplified by PCR and subsequently sequenced by Sanger sequencing as described before [12]. 

### 2.3. Phylogenetic Analysis to Evaluate Transmission Clustering Patterns

A concatenated alignment of NS5A and NS5B sequences was assembled for 89 samples; in two samples, sequencing was unsuccessful for the NS5A gene [14]. For each of these 89 included sequences, the 80 most highly similar HCV sequences were selected as control sequences from GenBank using the basic local alignment search tool (BLAST) [15]. After removal of duplicate sequences, phylogenetic analysis was reconstructed for 389 sequences, including, 87 study samples and 302 public sequences. We included a genotype 1b sequence (EU482849) to root the phylogenetic tree within the maximum likelihood (ML) framework. The ML tree was reconstructed using a generalized time-reversible nucleotide substitution model (GTR) with 4 discrete gamma rate categories, while accounting for invariable sites, as implemented within the IQTree software [16,17]. Tree robustness was evaluated using 1000 ultra-fast bootstrap replicates and a Shimodaira–Hasegawa-like (SH-like) approximate likelihood ratio test (aLRT) [18]. A cluster was defined based on a minimum ultra-fast bootstrap support of 90%, an aLRT of at least 80%, and a genetic distance threshold of 3% [19]. A transmission pair was defined as two sequences clustering together. The genetic distance was calculated using a Tamura and Nei model using the software MicrobeTrace [20]. Visualization of the clusters was also performed with the software MicrobeTrace. The robustness of our results was evaluated using an alignment of whole genome sequences (n = 44), and subsequently compared to an alignment of concatenated NS5A and NS5B sequences (n = 44) (Appendix A). 

## 3. Results

### 3.1. Study Population

From the 108 males with a genotype 1a HCV infection, 89 samples were available for sequencing. Sequences were generated, of which 44 with NGS (DAHHS 1 study) and 45 with Sanger (DAHHS 2 study). All males had MSM as the transmission route for their HCV infection and all had a recently acquired infection. Almost one fifth of patients (18%) originated from Belgium and others from several Dutch cities, with the majority from Amsterdam (33%) or Rotterdam (22%). Patients were all successfully treated with DAAs as part of the DAHHS 1 and 2 studies [11]. 

### 3.2. Recently Acquired HCV Infections in The Netherlands and Belgium Are Highly Clustered 

Phylogenetic analysis of sequences from recently acquired HCV infections showed that 78 of the 89 (88%) individuals included in the study cohort were embedded in one out of the seven identified clusters, and 4 of the 89 (4.5%) formed pairs (Figure 1 and Table 1). We identified four large clusters containing ≥10 individuals (depicted with the numbers II, III, V, and VI in Table 1) (Figure 1). 

All clusters showed a very strong geographical variation with mostly three different locations. Interestingly, all clusters, apart from one single cluster (number II in Figure 1), included samples from one of the two major Dutch cities (Amsterdam or Rotterdam) (Figure 1). 

Our study was not designed to study the HCV transmission dynamics between HIV-infected and HIV-uninfected MSM. Nevertheless, we found that the acute HCV samples of four HIV-negative MSM were embedded in clusters, which also consisted of acute HCV samples from HIV-positive MSM (Figure 1). Three clusters included HIV-negative MSM (depicted with number III, IV, and V in Table 1 and Figure 1) and all these individuals were sampled around the end of 2017 and originated from several different geographical locations. Interestingly, cluster IV included only three individuals, of which two were HIV-negative. 

### 3.3. Clusters Persist after Continuously Monitoring and Broadened Access of DAA Therapy in The Netherlands 

Although the widespread use of DAAs since 1st of November 2015 resulted in a profound reduction in the number of new HCV infections, little impact was found on the size of the phylogenetic clusters. We identified four clusters of phylogenetically linked HCV infections before the widespread introduction of DAAs in 2015 (number II, III, V, and VI in Table 1 and Figure 1), which were persistent after 2015. Moreover, a new cluster emerged after 2015 (number IV, VII, and VIII).

## 4. Discussion

This study describes substantial clustering of acute HCV infections in the Netherlands and Belgium. Over 90% of the recently acquired HCV infections were embedded in one of the detected clusters. HCV infections were detected in the years preceding, as well as after DAA therapy introduction, suggesting active transmission of HCV among HIV co-infected MSM. Importantly, this study showed that clusters present prior to the universal access of DAA therapy remained present, despite several elimination strategies, including compulsory notification of public health services of newly diagnosed HCV infections, partner notification and testing, annual HCV testing among HIV-positive MSM, and increased awareness amongst clinicians. Indeed, several Dutch studies show a stabilizing HCV incidence among HIV-infected MSM [21,22]. Our results confirm that unrestricted DAA therapy without other prevention methods led to a reduction in infections, however, will not suffice to micro-eliminate HCV among HIV-infected MSM [23,24]. 

Apart from persistent clustering, our study provided further insight into the transmission of recently acquired HCV infections, as we identified several clusters, of which four were major clusters with ten or more infections. As most of the recently acquired HCV infections were embedded in large clusters, the Dutch and Belgium HCV epidemics seem highly suitable for targeted prevention strategies. Targeted prevention strategies, such as behavioral interventions and intensified HCV testing in specific risk groups followed by immediate DAA treatment, could therefore be combined to stop HCV transmission in MSM [25,26].

Our data show geographical mixing not only between both neighboring countries, but also between the different Dutch centers. We identified that clusters are large with small genetic distances, and contain several Dutch cities, suggesting local transmission. In agreement with our finding, the Swiss HIV cohort showed an increase in domestic transmission during the period of 2008 and 2016 [27]. Nevertheless, we have indications that clusters could be part of European HCV networks for several reasons [10]. Firstly, we found clusters that embedded infections from Belgium and The Netherlands. Secondly, the largest cluster contained an M28V mutation in 96% of the cluster (described in S.Popping et al. Clin. Infect Diseases 2020) which is also found among HIV-positive MSM in Paris [14,28]. Lastly, data from van de Laar et al. show that samples from Dutch HIV-positive MSM are embedded in large European clusters [10]. Koopsen et al. describe a rise in the proportion of external introductions of new HCV infections in Amsterdam [29]. These findings, combined with our data, can suggest that Amsterdam is one of the entry points of HCV in the Netherlands. This is important information for trying to achieve HCV elimination, as this can guide testing programs and provide a more intensive approach in cities that are entry points. The use of real-time phylogenetic analysis combined with clinical and demographical information could help in identifying active transmission networks that should get appropriate attention from public health interventions. Moreover, further insights into the local and international transmission nature of new HCV infections are needed to further target elimination strategies. A collaboration with several large cities in Europe was initiated for further exploration.

In agreement with other findings, HIV-uninfected MSM diagnosed with a recently acquired HCV infection belonged to similar clusters also containing HIV-positive MSM [9,30,31]. This supports a model of an HCV spillover among high-risk MSM, regardless of their HIV-status. Importantly, among HIV-negative MSM, viral hepatitis screening is not standardized or regularly performed at sexual transmitted disease (STD) clinics. The actual HCV prevalence and incidence in this population is, therefore, unknown. Among HIV pre-exposure prophylaxis (PrEP) users in Amsterdam, a high baseline HCV prevalence of 4.8% was found, which was 2.9% in a similar cohort in Antwerp [7,8]. Although it is unclear whether these percentages can be extrapolated towards the HIV-negative MSM population with high-risk behavior not engaging in PrEP services. MSM PrEP users are currently monitored for HCV on a regular basis [32]. Moreover, prevention efforts should not be restricted to HIV status but towards risk behavior and thus be expanded towards the high-risk HIV-negative MSM population. 

This study has several limitations. Firstly, our controls were sequences submitted to GenBank, of which we know that the majority originated from samples of chronic HCV-infected patients rather than of acutely HCV-infected patients. Moreover, sequences included in GenBank are biased towards studies from countries with resources to perform genetic sequencing. Moreover, most available samples consist only of an NS5B fragment, as it is typically used for genotyping purposes. However, this is a very short stretch with a much lower phylogenetic signal as our concatenated alignment of the NS5A and NS5B genes.

Secondly, the HCV epidemic in the Netherlands is predominantly driven by MSM. New infections due to injection drug use became exceptional [33,34]. We do realize that, in several other countries, injection drug use remains the major driver of the HCV epidemic. However, interaction between MSM and injection drug users is highly unlikely. Additionally, needle sharing among MSM in the context of chemsex only rarely occurs [10,35]. 

In conclusion, we used well-defined data from the Netherlands and Belgium to unravel the transmission of recently acquired HCV infections among HIV-positive MSM. Our results indicate that recently acquired HCV infections among MSM in the Netherlands and Belgium are strongly clustered. Despite an HCV incidence reduction after high DAA uptake and continuous monitoring, HCV transmission continued to be present in the same clusters as beforehand. This shows that DAA therapy as a standalone intervention proved to be insufficient in obtaining micro-elimination, therefore, we advise to install additional prevention measurements. 

## Figures and Tables

**Figure 1 viruses-14-01953-f001:**
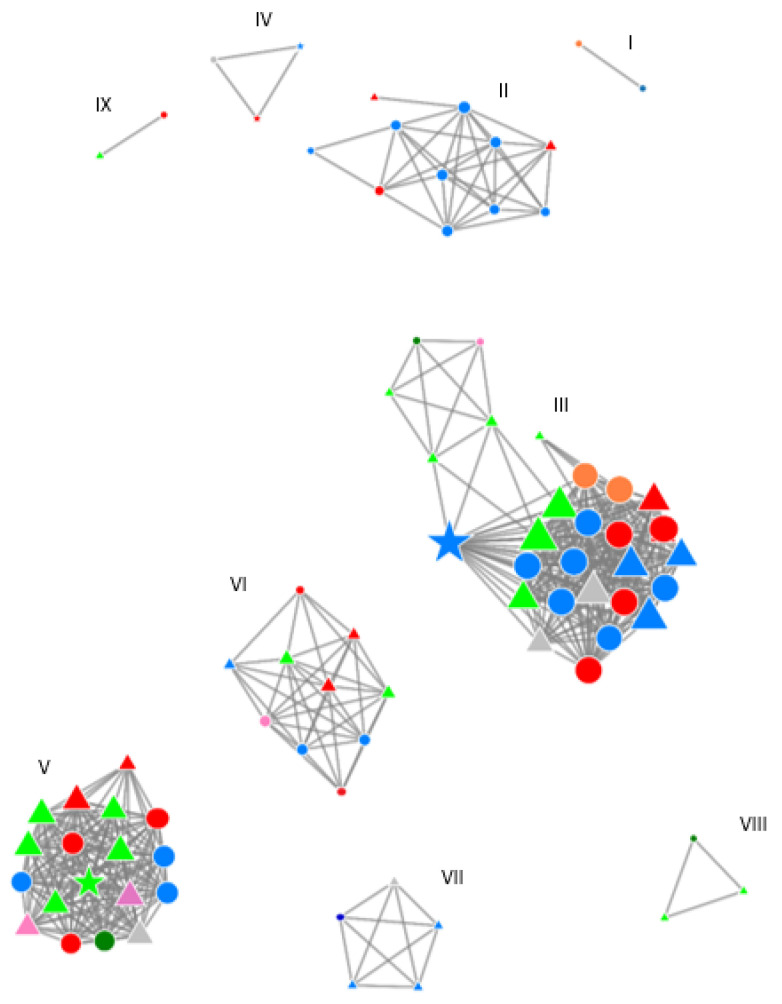
An overview of the clusters and transmission pairs (**I**–**VIII**) with a genetic distance threshold <3%. The circles represent HCV infections included prior to the widespread use of DAA therapy (all in HIV-positive MSM). The triangles represent HCV infections among HIV-positive and star among HIV-negative MSM after DAA introduction. The size of the nodes represents the number of sequences to which that particular node was phylogenetically linked. The colours represent the centres where patients were diagnosed with HCV (dark blue = Haarlem, dark green = The Hague, grey = Arnhem, light blue = Amsterdam, light green = Antwerp (Belgium), pink = Maastricht, orange = Utrecht, and red = Rotterdam) The clusters consisted only of study samples and did not include any of the control samples, which mostly originated from other European countries and the United States. The figure is established using Microbe Trace [20].

**Table 1 viruses-14-01953-t001:** An overview of the clusters and transmission pairs obtained with maximum likelihood phylogenetic reconstruction N = 82 sequences clustered in a cluster or pair. The other seven were not linked to any of the sequences in our set or reference set. The samples are divided in the ones included between 2013 and 2014 prior to the widespread use of the direct-acting antivirals in 2015 and 2016–2018.

Phylogenetic Cluster/Pair	I	II	III	IV	V	VI	VII	VIII	IX
Included 2013–2014	2	7	13	1	7	5	1	1	1
Included 2016–2018	0	3	15	2	11	5	4	2	1
Total number of MSM	2	10	28	3	18	10	5	3	2
Mean genetic distance	0.7	1.9	1.9	1.4	2.1	2.1	1.6	0.4	1.5

## Data Availability

Data is available upon request.

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
