# Peer review of "Persistent Transmission of HCV among Men Who Have Sex with Men despite Widespread Screening and Treatment with Direct-Acting Antivirals"

_viruses, 2022, doi:10.3390/v14091953_

Round 1

Reviewer 1 Report

The authors perfectly demonstrated the persistent transmission of HCV among MSM despite widespread screening and treatment with DAAs. The work was nicely done, and the findings are important. I have no further comments. 

Author Response

We thank the reviewer for the compliments.

Reviewer 2 Report

Congratulations to the authors for the work, it is very interesting and has a very important theme. However, some points need to be clarified and others should be taken with caution.

1- Lines 30-31: The sentence is not clear, it needs revision to make it more fluid.

2- Lines 34-35: But what was the difference between clusters? Specify.

3 - Attention to the citation format required by the journal.

4 - Lines 53-55: Is this a comment by the author? if not, it needs a reference.

5 - Line 72: How many individuals were assessed in total? How many samples were taken in total? This information is not very clear.

6 - Line 89: Why were only individuals with genotype 1a included?

7 - Lines 90-97: The explanation provided in the paragraph about the study two is not very clear.

8 - Lines 98-99: How many individuals had samples sequenced in total? How many for Sanger and how many for NGS? This information is not very clear.

9 - Table 1: The authors state that 87 sequences were included, why the sum of the table is only 82?

10 - Lines 181-184: This fact is not very clear with presented data.

11 - Lines 197-198: Where is the data to support this statement? This is not clearly presented in the results. Did these patients have any family, parental and/or sexual relationships between them?

12 - Lines 200-202: It would be interesting to present the amino acid substitution data in the results section to support the statement.

Reviewer 3 Report

Thank you for the opportunity to review this manuscript. It is a very interesting article. The Global Burden of Disease and other studies provide an increasingly precise description of the burden of viral hepatitis, which has been increasing since 1990. Chronic infections with HBV and HCV can both be treated with highly effective oral medicines.Despite high prices, many high-income countries have announced decisions to provide treatment for all persons infected with HCV, with minimal co-payments. In this manuscript they use phylogenetic analysis to provide further insight into HCV transmission among mostly HIV-infected MSM with a recently acquired HCV infection after DAAs became unrestrictedly available, in the Netherlands and Belgium.

However I have some suggestions.

The introduction could be implemented. It is necessary to provide some epidemiological information about HCV in Netherlands and Belgium.

In methods it is necessary to describe the study sample. In fact, the description of the study sample is missing. It is very important to add a table with this information.

Round 2

Reviewer 3 Report

I approve the new version of the article.